# Mucopolysaccharidosis: What Pediatric Rheumatologists and Orthopedics Need to Know

**DOI:** 10.3390/diagnostics13010075

**Published:** 2022-12-27

**Authors:** Stefania Costi, Roberto Felice Caporali, Achille Marino

**Affiliations:** 1Pediatric Rheumatology Unit, Gaetano Pini Hospital, 20122 Milan, Italy; 2Department of Clinical Sciences and Community Health, Research Center for Pediatric and Adult Rheumatic Diseases (RECAP.RD), University of Milan, 20122 Milan, Italy

**Keywords:** mucopolysaccharidosis, lysosomal storage diseases, early diagnosis, musculoskeletal, inflammation

## Abstract

Mucopolysaccharidosis (MPS) is a group of disorders caused by the reduced or absent activity of enzymes involved in the glycosaminoglycans (GAGs) degradation; the consequence is the progressive accumulation of the substrate (dermatan, heparan, keratan or chondroitin sulfate) in the lysosomes of cells belonging to several tissues. The rarity, the broad spectrum of manifestations, the lack of strict genotype-phenotype association, and the progressive nature of MPS make diagnosing this group of conditions challenging. Musculoskeletal involvement represents a common and prominent feature of MPS. Joint and bone abnormalities might be the main clue for diagnosing MPS, especially in attenuated phenotypes; therefore, it is essential to increase the awareness of these conditions among the pediatric rheumatology and orthopedic communities since early diagnosis and treatment are crucial to reduce the disease burden of these patients. Nowadays, enzyme replacement therapy (ERT) and hematopoietic stem cell transplantation (HSCT) are available for some MPS types. We describe the musculoskeletal characteristics of MPS patients through a literature review of MPS cases misdiagnosed as having rheumatologic or orthopedic conditions.

## 1. Introduction

Mucopolysaccharidosis (MPS) are inherited lysosomal storage diseases caused by the deficiency of enzymes necessary for the degradation of glycosaminoglycans (GAGs) such as dermatan, heparan, keratan or chondroitin sulphate [1]. The incomplete degradation of GAGs leads to accumulation of these substrates in lysosomes in different tissues and organs in the body resulting in multi-organ overload and dysfunction [1,2]. The severity can range from disability in early childhood to milder forms with late onset [3]. Depending on residual enzymatic activity, the same mutation may lead to different phenotypes. MPS are commonly grouped in neuronopathic and somatic forms, based on the central nervous system (CNS). The type of accumulated GAG affects clinical manifestations; for example, accumulation of heparan sulfate (HS) induces CNS manifestations, while accumulation of keratan sulfate (KS) and dermatan sulfate (DS) induces corneal opacities, bone abnormalities and heart disease without neurological impairment [4]. All MPS are transmitted with an autosomal recessive trait, except for MPS type II, which is X-Linked recessive transmitted [1]. Genetic and demographic features of MPS are summarized in Table 1. The global incidence is estimated around 3.5 per 100,000 live births [5], but the prevalence varies between different countries. Nowadays, the real incidence of MPS is difficult to assess, also due to frequent misdiagnosis. Indeed, in the first phases of the disease the clinical picture may be incomplete and the development of the child is frequently normal [6]. In this scenario, early diagnosis is mandatory because the effectiveness of therapy depends not only on the type of enzyme deficiency but also on the stage of the disease [7]. Early intervention can improve outcomes for MPS patients [8], especially for patients eligible for hematopoietic stem cell transplantation (HSCT) or enzyme replacement therapy (ERT).

The aim of this review is to elucidate the characteristics of MPS and offer an overview of the possible differential diagnosis from a rheumatological and orthopedic point of view.

## 2. Pathogenesis and the Role of Inflammation

GAGs are linear sulfated chains, typically associated with proteins to form proteoglycans and represent a crucial component of the extracellular matrix (ECM). GAGs not only have a structural function, but they are also involved in many cellular processes, including cells adhesion, signal transduction and activation of specific inflammatory pathways [9,10]. Dysfunction of these enzymes leads to the accumulation of GAGs in the lysosome, and in extracellular tissues affecting cellular homeostasis and cross-talk [11]. In MPS I-II and VII, the defective enzyme leads to accumulation of both HS and DS, while in MPS III only HS catabolism is involved [12]. HS chains act as a cofactor in chemokines-ligands binding and sulphation state [13,14], and chain length [15], which can be impaired in MPS [16], seems to play a key role in regulating these interactions. Conversely, DS seems to induce chondrocyte apoptosis directly [9]. Finally, the etiology of KS accumulation damage in MPS IV is very poorly understood. In recent years, the hypothesis of an inflammatory mediated state consequent to GAGs accumulation has been extensively studied, opening up new treatment options. First of all, autophagy is a process mediated by lysosomes. A study on MPS mice demonstrated impaired pH homeostasis in lysosomes with a lower content of H+ associated with an increased cytosolic Ca+ suggesting an increase in lysosome membrane permeability and a secondary dysfunction of autophagy [17,18]. In this setting, a secondary mitochondrial dysfunction may cause a release of reactive oxygen species and reactive nitrogen species producing oxidative stress (Figure 1) that can contribute to the inflammatory process [19,20,21,22,23]. In MPS I, the presence of HS in ECM with abnormal sulfation patterns may trigger the inflammation attracting leukocytes [24,25,26]. Another possible mechanism of inflammation is the stimulation of Toll-like receptors (TLRs) by GAGs. Indeed, in response to tissue injury, the degradation of hyaluronan GAG stimulates TLR4 signaling and mediates a potent inflammatory response activating dendritic cells (DCs) and macrophages [27,28]. In animal models, synovial fibroblasts and fluid showed elevated expression of numerous inflammatory molecules (i.e., tumor necrosis factor), including cytosolic pattern recognition receptors (i.e., TLR) [28]. TLR4 signaling leads to the activation of NF-kB pathway and a subsequent release of cytokines as tumor necrosis factor α (TNF-α) [29]. There are many pieces of evidence to support this theory, such as an increased expression of TLR4 gene in MPS IIIA mouse brain [30] or an increase in TNF-α release in mouse microglia culture added with GAGs from MPS IIIB patients [31]. In addition to TNF-α signaling, IL-1β and IL-6 seem to play a pivotal role in the pathogenesis of MPS [31]. It appears that IL-1β mainly participates in CNS inflammation, while TNF-α is primarily involved in musculoskeletal (MSK) manifestations [32]. IL-1β is not secreted in active form and its activation is mediated by inflammasome proteolytic cleavage [33]. Recently, a two-step model for inflammasome activation has been described. NF-kB pathway promotes NLR family pyrin domain containing 3 (NLRP3) inflammasome transcription (priming step) and other conditions as defective autophagy, lysosomal vacuolation, mitochondrial dysfunction and oxidative stress may mediate the NLRP3 inflammasome assembly and activation (activation step) (Figure 2) [32,34].

ERT is the only approved medical treatment for MPS that aims to reduce GAGs synthesis, however, it has no effect on inflammation. In MPS VI rats, treatment with infliximab prevented the elevation of TNF-α and NF-kB signaling not only in the blood but also in articular chondrocytes and fibroblast-like synoviocytes [35]. In another study on MPS VI rats, TNF-α inhibitor combined with ERT showed an improvement in MSK outcome (particularly in motor activity and mobility) [36].

## 3. Musculoskeletal Manifestations in MPS

Even with a broad phenotypic spectrum, the involvement of bone, cartilage, ligaments, tendons, joint capsules, and all the soft tissues near the joints represents a common feature among all MPS types [6,37]. The timing and severity of such involvement are unpredictable, as in other organs impaired by GAG infiltration. Table 2 shows the most common MSK manifestations in MPS. Patients with attenuated forms of MPS might first seek medical attention for MSK complaints; thus, pediatric rheumatologists and orthopedic surgeons might be consulted as the first healthcare specialist. Unfortunately, mild MPS subjects might be frequently misdiagnosed as having other diseases such as juvenile idiopathic arthritis, osteochondrodysplasias, or other disorders (Table 3) [3,38,39,40,41].

### 3.1. Arthropathy and related issues

#### 3.1.1. Joint Stiffness and Contractures

The impaired skeletal remodeling and the GAG infiltration of tissues around and within the joints are the causes of joint stiffness and contracture in all but one type of MPS (hypermobility is characteristic of MPS IV) [1,6,37]. Joint stiffness and contracture in MPS are progressive and present a symmetrical distribution with all joints potentially affected [3]. In MPS patients, the local and systemic signs of inflammation are absent (warmth, swelling, tenderness, fever, increased inflammatory markers). The classical claw hand deformity observed in MPS patients is due to skeletal abnormalities and contracture of interphalangeal (IP) joints with distal IP (DIP) more frequently involved. Furthermore, the bone enlargement near the joints in MPS may result in a swollen appearance mimicking arthritis. In this setting, juvenile idiopathic arthritis (JIA) represents a common differential diagnosis; however, several clues might help clinicians properly differentiate these two conditions (Table 4). Ultrasonography represents a useful tool in joint assessment of these patients since it can detect abnormal intraarticular material with peri-synovial Doppler signal but without effusions and a clear distribution to synovial recesses; flexor and extensor tendons of fingers might show a normal structure, whereas retinacula and flexor tendon pulleys might be thickened [42]. Sometimes the skin over the involved joint might become tight and thick, resembling the skin of scleroderma patients [37]. Alongside congenital arthrogryposis that is thought to result from decreased intrauterine movement [43], two other conditions might mimic the findings of MPS: camptodactyly and cheiroarthropathy. The flexion deformities of the proximal IP (PIP) joints of camptodactyly can be found as isolated features or as part of a syndrome such as camptodactyly-arthropathy-coxa vara-pericarditis (CACP) [43]. The pattern of affected joints (PIP in camptodactyly and DIP in MPS), and the involvement extension (the fifth finger in camptodactyly, whereas all fingers can be progressively affected in MPS) allow for discriminating between MPS and camptodactyly. Furthermore, dysostosis multiplex and other extra-articular features are absent in CACP and other conditions associated with camptodactyly. Cheiroarthropathy is typically associated with type I and type II diabetes; in this condition, PIP joints of the fourth and fifth fingers are involved, even though all the joints of the hand might be affected later on. These changes might be accompanied by skin tightening and thickening resembling scleroderma; indeed, cheiroarthropathy has been previously described as pseudoscleroderma [44]. The simultaneous presence of several MSK abnormalities (pes cavus, metatarsus adductus and equinovarus deformities, hip dysplasia, genu valgum) can affect the patient’s ability to walk [6,45]. Toe-walking can be observed in MPS patients when ankles and Achilles tendons are involved.

#### 3.1.2. Joint Stiffness and Contractures

MPS IV (Morquio syndrome) is the only MPS type characterized by the presence of joint hypermobility due to several abnormalities of periarticular connective tissues and bones (bone hypoplasia and metaphyseal deformity) [46]. Proximal stiffness and distal hypermobility of the joints are the typical findings in MPS IV that may impact daily activity (e.g., dressing and personal care). Furthermore, these patients present a high risk of atlanto-axial subluxation caused by the concomitant presence of joint hypermobility and odontoid hypoplasia [37,47,48].

#### 3.1.3. Trigger Digits and Carpal Tunnel Syndrome (CTS)

The GAG deposition in the flexor tendons and capsular tissues is responsible for the trigger finger. Triggering is usually a clinical diagnosis made upon the usual locking and catching symptoms along with ultrasound findings that help understand the underlying cause [49,50]. Indeed, inflammatory tenosynovitis may also provoke a discrepancy between the flexor tendon sheath’s size and the enclosing fibro-osseous canal, ultimately resulting in a trigger digit [51]. Since CTS is rare in children, the presence of this uncommon sign should raise the suspicion of a possible MPS, especially when associated with a trigger finger [52]. The thickening of the flexor retinaculum and tendon sheaths due to GAG deposition causes median nerve compression [53]. The early diagnosis of CTS in MPS patients might be challenging given the lack of typical complaints (numbness or pain) partially due to intellectual and verbal limitations of these subjects [3,54,55]. Data coming from an international registry on MPS I confirmed CTS as a common manifestation (30%); however, in the attenuated form, the latency between the CTS and MPS diagnosis resulted in more than 7 years [56]. Other experiences confirmed the CTS as common finding in mild diseases. In a case series reporting thirteen patients with attenuated MPS I, ten patients presented CTS, with two of them having concomitant triggering [3]. In another study, the electromyographic or nerve conduction velocity testing was used to diagnose CTS in 22 patients with MPS. Of the 17 subjects with CTS, 8 children (45 fingers) had trigger digits too [55].

### 3.2. Skeletal Involvement

The skeletal alterations seen in MPS and the related radiographic changes are regrouped under the term result from impairment in endochondral and membranous bone growth [9]. All bones of the body can be affected. Usually, long bones of the extremities show thickened diaphysis and hypoplastic epiphyses. The Madelung’s deformity with the distal radius curved towards a hypoplastic ulna, and the bullet shape metacarpals (short and thin with proximal tapering) are the typical alteration of upper limbs. Other abnormalities include thick calvarium, j-shaped sella turcica, enlargement of the skull, short and thickened clavicles, and oar-shaped ribs [37,45,57,58].

#### 3.2.1. Kyphosis

Kyphosis might occur in all MPS types but is almost always present in severe forms of MPS I [59,60]. The kyphotic deformity is often due to hypoplastic vertebral bodies, especially in the anterior-superior part, and can be associated with scoliosis [45]. Even after hematopoietic stem cell transplant (HSCT), the progression of kyphosis often requires a surgical intervention despite conservative treatments such as strengthening activities and physical therapy [57]. It is reasonable to postpone surgery as long as possible both to allow for the maximum growth of the spine and for the consideration of the high risk of these procedures in MPS. However, if signs of myelopathy are detected, invasive procedures cannot be delayed. Young children who are not eligible for surgery may benefit from bracing [59].

#### 3.2.2. Hip Dysplasia

Hip dysplasia in MPS comes from the variable combination of the flattened acetabulum, hypoplasia of the proximal epiphysis in its medial portion, and coxa valga. Therefore, children may develop several dysfunctions and limitations due to progressive instability and a higher risk of dislocation. Hip dysplasia affects several types of MPS (MPS I, MPS II, MPS III, MPS IV, and MPS VI), occurring in almost all subjects with Hurler syndrome (severe MPS I) [6]. Hip pain has been observed as presenting manifestation of MPS, and it can be easily misinterpreted [40,61]. In this regard, ultrasound may help in the diagnosis along with the whole clinical phenotype [62]. Hip surgery is deemed necessary in Hurler syndrome patients given its severity; on the contrary, it is not recommended for MPS III and IV due to the risk of osteonecrosis of the femur head [63,64,65].

#### 3.2.3. Genu Valgum

Knee deformity is usually severe in MPS I, MPS IV, and MPS VI requiring surgery [46,66,67]. Total knee arthroplasty is necessary for adult MPS patients with severe arthrosis due to impairment of articular cartilage [68]. Eight-plate hemiepiphysiodeses at the distal femoral and/or proximal tibial physis were performed in 23 patients with MPS IV progressive or severe genu valgum with a mean follow-up of 44 months. The mean age at surgery was 8.3 years. Outcome measures documented a significative intermalleolar distance reduction without major complications (infections, implant failures, or loosening); three patients required a repeated procedure after a mean 22 months after plate removal [67]. Compared to osteotomy, hemiepiphysiodesis is less invasive and has a lower mobility impact.

#### 3.2.4. Spinal Cord Compression

Hypoplasia of the odontoid process is frequently observed in MPS, especially in Hurler syndrome and Morquio syndrome. Odontoid hypoplasia is the leading cause of atlanto-axial instability predisposing to subluxation that may result in spinal cord compression in combination with GAG infiltration of the dura [69,70,71]. The resulting development of spastic tetraparesis is the most common finding; however, hemiparesis and paraparesis have also been documented [70,71]. HSCT and enzyme replacement therapy (ERT) seem to have a positive impact on spinal cord compression; however, larger studies are needed to verify this opportunity [72,73].

#### 3.2.5. Consideration on Surgery in MPS

The treatment approach for skeletal abnormalities in MPS is mainly based on individual cases. However, severe and progressive deformities that may have serious consequences (e.g., mobility limitation, spinal cord compression) often require invasive procedures [69,74]. Patients with MPS have high risk of surgery for many reasons. Before surgical interventions, physicians should carefully assess spinal alterations such as neck stiffness, atlanto-axial instability, and cervical/occipital stenosis. Furthermore, endotracheal intubation can be challenging for these patients for several reasons: macroglossia, hypertrophy of the adenoids and tonsils, reduced mouth opening, and increased secretion. Fiberoptic intubation through a supraglottic airway seems associated with the lowest risk of perioperative adverse events and the lowest need for the transition to a rescue airway technique [75]. Other risks might lie in restrictive pulmonary disease, hypoplastic cartilages of the trachea, and cardiac involvement (mild to severe valvulitis and cardiomyopathy). Therefore, such patients need an accurate pre-operative assessment as well as a close post-operative observation also for minor procedures [45,57].

#### 3.2.6. Short Stature

Growth impairment is a frequent manifestation of all MPS types and is caused by the alteration of the scheduled chondrocyte maturation in the growth plate [76]. The result is a non-harmonic short stature with axial development more compromised than appendicular growth [77]. The efficacy of recombinant growth hormone in these patients is variable, with a high rate of unresponsiveness. Final height can be severely impaired even though attenuated types might have a subtle decrease in growth velocity or even a normal maturation [78]. The slowing down of growth in untreated MPS I patients starts near 2 years of age and usually becomes evident by the age of 9 (below the third percentile) [79,80]. Data coming from the MPS I Registry documented a slower decline in height z-scores during laronidase ERT period compared to the natural history interval; however, the median height remained below normal standards overall [80,81]. An algorithm for early diagnosis of attenuated MPS I based on the primary observation of growth impairment in children above 5 years of age has been proposed [82].

### 3.3. Musculoskeletal Biomarkers

Nowadays, there are not validated biomarkers able to predict the progression of the disease or the response to therapy. TNF-α levels, regardless of ERT therapy, are higher in patients suffering from MPS when compared with healthy controls, and they are associated with MSK symptoms and limitations. However, elevated levels of TNF-α appear not to correlate with general health score [29]. In MPS, inflammation and direct chondrocyte apoptosis mediated by KS lead to osteoclasts, osteoblasts and chondrocytes dysfunction resulting in abnormal bone matrix deposition and failure of endochondral ossification [83]. In MPS IVA, KS accumulates in cartilage, leading to specific skeletal abnormalities. In these patients KS, KS sulfation levels, chondroitin-6-sulfate levels, and collagen type II are potential biomarkers associated with bone and cartilage disease [84]. Increased osteoblast activity may be suggested by elevated levels of osteocalcin in MPS patients [85]. In a study on MPS VII dogs, bone formation markers, such as alkaline phosphatase, osteopontin, and osteocalcin were lower in MPS than in controls suggesting a delay in bone formation [86].

On the other hand, elevated levels of IL-6 seem to predict progression in joint contracture, short stature, and hip dysplasia [87]. Simonaro et al. reported chondrocyte apoptosis in the articular cartilage of rats and cats with MPS VI. They found increased levels of IL-1, TNF- α, and nitric oxide in chondrocyte cultures of MPS animal model compared with normal cells [88]. A recent study on MPS I patients revealed elevated levels of IL-1β, TNF-α, osteocalcin, pyridinolines (PYD), and deoxypyridinolines in MPS patients compared to healthy controls. In addition, IL-6 and PYD levels appear to be associated with progression in joint contracture, short stature, and hip dysplasia [89]. Although the wide heterogeneity of MPS makes finding reliable markers challenging, cytokines and molecules related to the degradation of the ECM might be candidates of MSK biomarkers. Further studies are needed to validate specific and sensitive markers that can help in clinical practice.

### 3.4. Bone Status in MPS

Patients with MPS are theoretically at risk of low bone mass density (BMD) due to limited mobility secondary to bone dysplasia and contractures. The development of bone mass in children is a dynamic process of adapting to mechanical stimuli. Physical exercise is necessary to stimulate bone cells in order to reinforce bone structure and promote bone remodeling. Fractures after low energy trauma have been described in MPS patients but their incidence is low [90,91,92,93]. In a study on 40 MPS subjects, the prevalence of low BMD was 48% for lumbar vertebra, but the rate of DXA z-score < −2 reduced to 6% after correction for height-for-age z-score (HAZ). Indeed, a high percentage of patients with MPS present with short stature; therefore, DXA should be evaluated by adjusting for height (HAZ) [94]. A recent study on 126 MPS patients of various types found a short stature in 67.5%. Furthermore, 13.5% of patients were immobile, and 28.6% had 25(OH)D3 deficiency. In this cohort, BMD z score was < −2 in 40% of patients. However, after HAZ adjustment, only 2.2% had a BMD z score < −2 [95]. These data suggest the prevalence of low BMD in patients with MPS is probably overestimated and correction for HAZ is necessary due to abnormal bone development in MPS.

## 4. Other Manifestations of Attenuated MPS

The correlation between underlying mutations and residual enzymatic activity causes the broad phenotypic spectrum, impacting the timing of the manifestations’ development [1,3]. The inter- and intra-phenotypic variability of MPS makes the diagnosis of MPS challenging, especially for attenuated forms. The lack of neurological involvement along with the absence of manifest dysmorphisms such as coarse face and evident skeletal abnormalities may partially explain the frequent diagnosis delay [3]. Subtle disease progression and unspecific early symptoms might be unnoticed by healthcare providers unfamiliar with MPS. Alongside MSK involvement, late-onset MPS patients more frequently have eye- and heart-related issues [39]. Non MKS features are summarized in Table 5.

The most common ocular finding in MPS is corneal clouding which may severely impact visual acuity. It is due to the deposition of GAG granules with a yellowish-grey color in all corneal layers. MPS I and MPS VI have a more severe manifestation than MPS IV and MPS VII; on the other hand, patients with MPS II and III might present very mild corneal involvement [96]. Corneal transplantation is often needed in early adult patients; however, the concomitant presence of retinopathy represents a poor prognostic factor [97,98]. Glaucoma, increased intraocular pressure, optical nerve abnormalities, and retinal degeneration might be associated with corneal clouding, which usually begins around 10 years of age in attenuated MPS I [96,97,99,100]. Patients with MPS I, II, and III are more prone to develop retinal degeneration in adulthood, which is rarely seen in MPS IV [96].

Heart involvement is a common feature across all MPS types representing a significant cause of morbidity and mortality [101,102]. Patients with mild form frequently show valvular disease (thickening, regurgitation and/or stenosis) with mitral and aortic valves commonly affected [103]. The rapidity of the valvular disease progression is influenced by the accumulated GAG; indeed, those diseases characterized by dermatan and heparan sulfate deposition (MPS I, MPS II, and MPS VI) show a severe course of valvulopathy compared to types in which there is no such infiltration (MPS III and MPS IV) [103,104]. GAG storage may also cause left ventricular hypertrophy, coronary artery disease, abnormal diastolic function, arrhythmia, and complete heart block [104,105]. Furthermore, airways obstruction, rib deformity, and involvement of pulmonary vessels may result in pulmonary hypertension. ERT might slower down the valvular disease and ameliorate the ventricular hypertrophy [105,106].

Except for MPS III, cognitive impairment is uncommonly seen in the early stage of attenuated forms. Nevertheless, MPS I and II patients may develop variable degrees of intellectual disabilities during the disease course [39,107].

Among the other possible manifestations, the respiratory system is frequently involved (rhinosinusitis, otitis media, restrictive lung disease, sleep apnea, snoring) due to the GAG submucosal deposition causing airway obstruction, potentially, in every part of the respiratory tract [108]. Tracheomalacia, resulting in tracheal stenosis, might be life-threatening, especially in MPS II and IV, where this complication is more frequent. The first signs may arise during adolescence with further adult progression [109]. The GAG storage in the ear system causes conductive or sensorineural hearing loss, typically in the high-frequency range. Inguinal and umbilical hernias are quite common, especially in attenuated MPS I (till 65% of cases), and often may precede the onset of MSK manifestations [99,110]. Several degrees of hepatosplenomegaly and diarrhea are frequently observed in MPS [100,111].

## 5. Diagnosis

The accumulation of undegraded GAGs increases these substrates in urine, blood [112,113], and cerebral spinal fluid [114,115]. In children, the first best step in management is measuring the levels of urinary GAGs, as it is an excellent test in terms of sensitivity and specificity [116]. Quantitative analysis allows for the identification of an abnormal accumulation of urinary GAGs, while qualitative (electrophoresis or chromatography) analysis determines the type of accumulated GAGs. It is recommended that the first morning urine is collected because they are concentrated in order to avoid false negative results [117]. An elevated GAG concentration in urine is highly suggestive of MPS, but a negative test does not rule out the diagnosis [118]; in fact, a false negative test may occur in MPS III and IV [119,120] due to GAGs concentration declining with age. Definitive diagnosis requires enzyme activity assays, usually tested on peripheral blood leukocytes, and gene sequencing is necessary for identifying the underling mutation.

Prenatal diagnosis is based on measuring the enzyme activity in cultivated chorionic villi or amniocytes in case of a family history of MPS or hydrops fetalis. Testing for MPS I has been included in the United States newborn screening panel since 2016, and it is currently underway in other countries [121,122].

## 6. Conclusions

The suspicion of a possible diagnosis of MPS should rise based on the concomitant presence of some of the abovementioned manifestations. However, physicians may focus their attention to symptoms related to each own expertise and overlook other possible hints. In this regard the anamnesis plays a crucial role; a careful medical history of the patient and the family may reveal unexpected clues. A semi-structured medical history form has been proposed to help with medical interviews of suspected MPS patients [109]. Furthermore, the Cimaz algorithm, despite its countless variations, remains a milestone for the diagnosis of attenuated MPS with joint involvement [123]. MPS manifestations might be overlooked, carrying the risk of misdiagnosis. Therefore, awareness of MPS phenotypes should be raised, particularly among pediatric rheumatologists and orthopedic surgeons, since MSK involvement is common in attenuated forms. Alongside progressive joint contracture without inflammation, looking for other features in the medical history might avoid mistakes and delays in diagnosis. Once the suspicion of MPS is raised, urinary GAG is recommended, with enzyme assay and genetic testing as the following tests if an abnormal pattern is found.

## Figures and Tables

**Figure 1 diagnostics-13-00075-f001:**
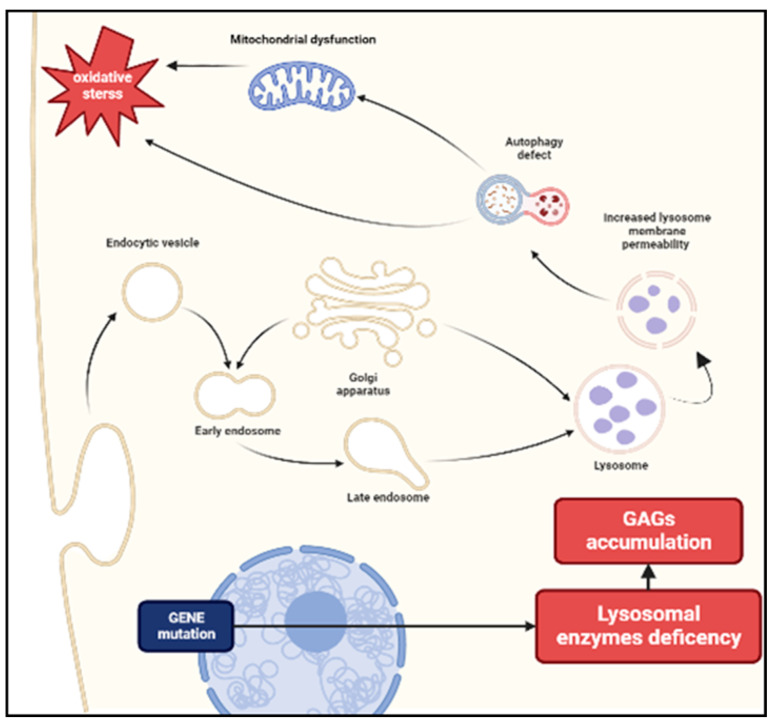
Pathogenesis of MPS. The deficiency of enzymes necessary for the degradation of glycosaminoglycans (GAGs) leads to an accumulation of these substrates in lysosomes. Increased lysosomal membrane permeability influences cellular homeostasis and causes impaired vesicular fusion and autophagy. A secondary mitochondrial dysfunction may cause a release of reactive oxygen species resulting in oxidative stress.

**Figure 2 diagnostics-13-00075-f002:**
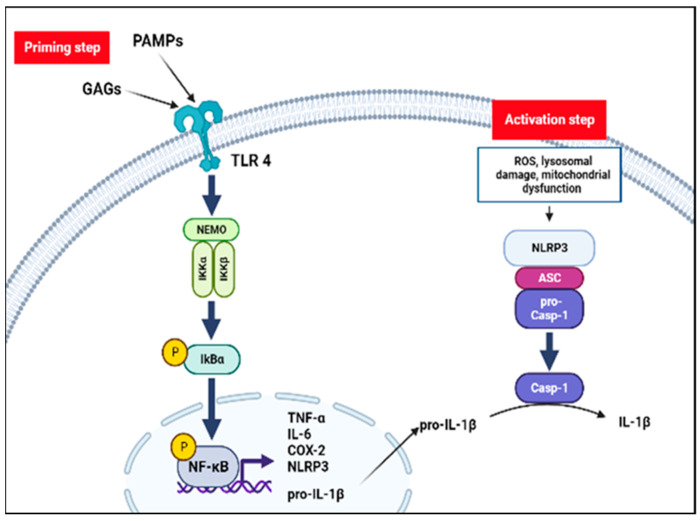
The inflammatory hypothesis in MPS. Inflammasome activation requires two-step. Pathogen-associated molecular-patterns (PAMPs) and unfractionated glycosaminoglycans (GAGs) stimulate inflammatory processes through a Toll-like receptor 4 (TLR4) pathway: the inhibitory subunit of NF-kB alpha (IκBα) is phosphorylated by the NF-kappa-B essential modulator (NEMO) complex liberating NF-κB dimer. Translocation of NF-kB dimer to the nucleus promotes NOD-like receptors family pyrin domain containing 3 (NLRP3) and proinflammatory cytokines transcription (priming step). Lysosomal damage, mitochondrial dysfunction and oxidative stress may mediate the NLRP3 inflammasome assembly and activation (activation step). NLRP3 together with the adaptor apoptosis-associated speck-like protein containing a CARD (ASC) protein promotes the activation of caspase-1 and forms NLRP3 inflammasome complex. Finally, caspase-1 mediates conversion to the active form of IL-1β.

**Table 1 diagnostics-13-00075-t001:** Genetical and demographical features of MPS.

Disease	OMIM	Enzyme Deficiency	Gene (Locus)	Inheritance	Incidence (1/100,000 Live Births)	GAG	CNS	Approved Therapy
**MPS I**		α-L-Iduronidase	IDUA (4p16.3)	AR	0.69–1.66	Dermatan sulfate, heparan sulfate	+	Laronidase
Type Hurler	607,014						+++	
Type Scheie	607,016						−	
Type Hurler/Scheie	607,016						+/−	
**MPS II** (Hunter syndrome)	309,900	Iduronate sulphate sulphatase	IDS (Xq28)	X-Linked recessive	0.3–0.71	Dermatan sulfate, heparan sulfate	+	Idrosulfase
**MPS III (Sanfilippo syndrome)**				AR		Heparan sulfate	+++	NA
Type III-A	252,900	Heparan-S-sulphate-sulphaminidase	SGSH (17q25.3)		0.29–1.89			
Type III-B	252,920	N-Acetyl-D-glucosaminidase	NAGLU (17q21.2)		0.42–0.72			
Type III-C	252,930	Acetyl-CoA-glucosaminide-N-acetyltransferase	HGSNAT (8p11.21)		0.07–0.21			
Type III-D	252,940	N-Acetylglucosaminine-6-sulphate-sulphatase	GNS (12q14.3)		0.1			
**MPS IV (Morquio syndrome)**				AR	0.2–1.3		−	
Type IV-A	253,000	Galactosamine-6-sulphate-sulphatase	GALNS (16q24.3)		0.2–1.3	A: Keratan sulfate, chondroitin sulfate		Elosulfase alpha
Type IV-B	253,010	β-galactosidase	GALNS (16q24.3)		0.02–0.14	B: Keratan sulfate		
**MPS VI** (Maroteaux–Lamy syndrome)	253,200	Arylsulfatase B	ARSB (5q14.1)	AR	0.36–1.3	Dermatan sulfate, heparan sulfate	−	Galsulfase
**MPS VII** (Sly syndrome)	253,220	β-Glucuronidase	GUSB (7q11.21)	Automosal recessive	0.05–0.29	Dermatan sulfate, heparan sulfate, chondroitin sulfate	+/−	Vestronidase alpha
**MPS IX** (Hyaluronidase deficiency)	601,492	Hyaluronidase		Automosal recessive	<0.01	Hyaluronan	−	NA

MPS = Mucopolysaccharidosis; OMIM = Online Mendelian Inheritance in Man; CNS = central nervous system; GAG = glycosaminoglycans; AR = autosomal recessive; NA = not available.

**Table 2 diagnostics-13-00075-t002:** Musculoskeletal features of MPS.

MPS Type	Musculoskeletal Features
MPS I (Hurler, Hruler-Scheie, Scheie)	Dysostosis multiplex, short stature (disproportionate), joint contractures, carpal tunnel syndrome, trigger digits, odontoid hypoplasia, atlanto-axial instability, acetabular dysplasia, coxa valga, genu valgum
MPS II (Hunter)	Dysostosis multiplex, short stature (disproportionate), joint contractures, carpal tunnel syndrome, trigger digits, odontoid hypoplasia, atlanto-axial instability, acetabular dysplasia, coxa valga, genu valgum
MPS III (Sanfilippo)	Mild short stature and contractures (mainly elbow joint)
MPS IV (Morquio)	Severe skeletal dysplasia, dysostosis multiplex, short stature (disproportionate), joint hypermobility, odontoid hypoplasia, atlanto-axial instability, acetabular dysplasia, hip dislocations, coxa valga, genu valgum, pes planus, pectus carinatum
MPS VI (Maroteaux-Lamy)	Dysostosis multiplex, short stature (disproportionate), joint contractures, carpal tunnel syndrome, odontoid hypoplasia, atlanto-axial instability, acetabular dysplasia, coxa valga, genu valgum, trigger digits, pectus carinatum
MPS VII (Sly)	Dysostosis multiplex, short stature (disproportionate), joint contractures, odontoid hypoplasia, atlanto-axial instability, acetabular dysplasia, pectus carinatum
MPS IX (Hyaluronidase deficiency)	Short stature, periarticular soft tissue masses, nodular synovial masses, joint effusions, acetabular erosions

**Table 3 diagnostics-13-00075-t003:** Differential diagnosis of musculoskeletal manifestations in MPS.

**Inflammatory diseases**	Inflammatory arthritis Scleroderma Dermatomyositis and polymyositis
**Distal extremity conditions**	Arthrogryposis Camptodactyly Clinodactyly Trigger finger (isolated) Carpal tunnel syndrome (isolated)
**Osteochondrodysplasias**	Epiphyseal dysplasia Spondyloepiphyseal dysplasia congenital (including the X-linked form) Spondylometaphyseal dysplasia Dystrophic dysplasias Osteogenesis imperfecta Other dysplasias
**Other metabolic diseases**	Gaucher’s disease Fabry’s disease Pompe’s disease Rickets Hypophosphatasia Diabetic cheiroarthropathy
**Miscellaneous**	Legg-Perthes-Calvé disease Growing pains Amplified musculoskeletal pain syndrome (AMPS) Muscular dystrophy Polyneuropathy Ehler-Danlos syndrome

**Table 4 diagnostics-13-00075-t004:** MPS arthropathy vs. Juvenile idiopathic arthritis (JIA).

Features of Joint Involvement	MPS	JIA
Involved joints	DIP	PIP and MCP
Stiffness temporal pattern	Continuous	Morning ^1^
Clinical signs	Stiffness and contracture ^2^	Joint swelling, warmth, and tenderness
Inflammatory markers	Normal	Normal/raised
Response to anti-inflammatory drugs ^3^	No	Yes

DIP: distal interphalangeal joints; PIP: proximal interphalangeal joints, MCP: metacarpophalangeal joints; ^1^ usually worse in the morning, exacerbated by rest, and relieved by activity; ^2^ some joints may have a swollen appearance, caused by underlying bony enlargement rather than synovial effusion; ^3^ steroidal and non-steroidal anti-inflammatory drugs.

**Table 5 diagnostics-13-00075-t005:** Non musculoskeletal features of different type of MPS.

	Neurological	ENT	GI	Cardiological	Ophthalmological	Hepato-Splenomegaly	Skin Involvement
MPS I H	Hydrocephalus Psychomotor retardation Behavior trouble Peripheral compression Atlanto-axial instability	Deafness (+++ H) Recurrent sinopulmonary infections Chronic rhinitis	Umbilical or inguinal hernias	Valve disease Cardiomyopathy Endocardial fibroelastosis Coronary heart disease	Corneal clouding Retinopathy Optical nerve compression	++	Thickened and rough skin texture Pebbly papules (rare)
MPS I S	-		-			++	
MPS I HS	Pachymeningitis cervicalis Typically normal intelligence		Umbilical or inguinal hernias			++	
MPS II	Neurocognitive decline Behavior trouble Some patients have normal intelligence	Deafness	-	Valve disease Cardiomyopathy Endocardial fibroelastosis Coronary heart disease	-	++	Pebbly papules
MPS III	Neurocognitive decline Behavior trouble Intellectual disability	Recurrent sinopulmonary infections Deafness	Umbilical or inguinal hernias	Milder forms of valve disease and Cardiomyopathy	-	+	Thickened and rough skin texture
MPS IV	-	Recurrent sinopulmonary infections Deafness	-	Milder forms of valve disease and Cardiomyopathy	Mild corneal opacities	+	Thickened and rough skin texture
MPS VI	Pachymeningitis cervicalis with normal intelligence	Recurrent sinopulmonary infections OSAS Pulmonary hypertension	-	Valve disease Cardiomyopathy Endocardial fibroelastosis Coronary heart disease	Corneal clouding	++	Thickened and rough skin texture
MPS VII	Hydrops fetalis Intellectual disability (mild or absent)	++	-	Milder forms of valve disease and Cardiomyopathy	Corneal clouding Retinopathy Optical nerve compression	+	Thickened and rough skin texture

MPS = mucopolysaccharidoses; H = Hurler; S = Scheie; HS = Hurler-Scheie; GI = gastrointestinal; ENT = ear nose throat; OSAS = obstructive sleep apnea syndrome.

## Data Availability

Not applicable.

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
