# Peer review of "Mucopolysaccharidosis: What Pediatric Rheumatologists and Orthopedics Need to Know"

_diagnostics, 2022, doi:10.3390/diagnostics13010075_

Round 1

Reviewer 1 Report

This paper nicely reviews the musculoskeletal involvement in various MPSs with a particular focus on differential diagnosis, mainly involving late onset cases which usually present with more subtle and unspecific findings.

The manuscript is well written; however, I have a couple of suggestions:

Line 40: change Genetical and demographical features with genetic and demographic features

Line 44: change the clinic may to the clinical picture may

Line 69-70:change accumulation of GAGs in lysosome, to accumulation of GAGs in the lysosome

Author Response

Thank you for the comments; we modified the text as suggested (see line 40, line 44, and lines 69-70). 

The professional revision was made through Grammarly premium. Here is the self-certification of English revision:

Dear Editors of Diagnostic, 
as corresponding author, I declare to have extensively revised the manuscript entitled "Mucopolysaccharidosis: what pediatric rheumatologists and orthopedics need to know." for English issues, according to the following parameters:
- Formal Language 
- Grammar (correct use of tenses)
- Spellings
- Punctuation/prepositions/articles and typographical errors - No vague content (Clarity of expressions)
- Correct presentation of ideas, facts and logic. 
The professional revision was made through Grammarly premium (https://www.grammarly.com/premium). An official certificate is not available, but I certify to meet all the requirements. 

Reviewer 2 Report

The authors make a good job revieweing the main muskoskeletal abnormalities in MPS patients. This is focused to rheumatologists and orthopedics.

The manuscript is well written, I would only suggest minor comments:

a) In the growth section- I would suggest to include that the patients are not responsive to treatment with growth Hormone

b) There are some typo mistakes in the manuscript, such as in table 1 Table 1- "Vestronidase alfa" is written as "Vestronidase alfavjbk". Please revise the final version of the text.

c) the authors call  nervous central system (NCS). I would suggest to change for central nervous system (SNC)

Author Response

Glad about your suggestions. 

  1. We added the following sentence (line 395): “The efficacy of recombinant growth hormone in these patients is variable with a high rate of unresponsiveness.”
  2. Thank you, we revised the manuscript and modified it accordingly. The professional revision was made through Grammarly premium. Here is the self-certification of English revision:

Dear Editors of Diagnostic, 
as corresponding author, I declare to have extensively revised the manuscript entitled "Mucopolysaccharidosis: what pediatric rheumatologists and orthopedics need to know." for English issues, according to the following parameters:
- Formal Language 
- Grammar (correct use of tenses)
- Spellings
- Punctuation/prepositions/articles and typographical errors - No vague content (Clarity of expressions)
- Correct presentation of ideas, facts and logic. 
The professional revision was made through Grammarly premium (https://www.grammarly.com/premium). An official certificate is not available, but I certify to meet all the requirements. 

3. We edited the text using “central nervous system (CNS).”